# Evaluation of Changes in the Patency of the Nasal Cavity and Eustachian Tube Depending on the Phase of the Menstrual Cycle: A Pilot Study

**DOI:** 10.3390/diagnostics14182044

**Published:** 2024-09-14

**Authors:** Alicja Grajczyk, Krystyna Sobczyk, Karolina Dżaman

**Affiliations:** Department of Otolaryngology, Centre of Postgraduate Medical Education, Marymoncka 99/103, 01-813 Warsaw, Poland; alicja.grajczyk@gmail.com (A.G.); krsobczyk@gmail.com (K.S.)

**Keywords:** menstrual cycle, luteal phase, follicular phase, rhinomanometry, tympanometry

## Abstract

(1) Background: Estrogen and progesterone, hormones specific to females, undergo fluctuations during the menstrual cycle. The aim of this study was to assess subjective and objective changes in nasal cavity and Eustachian tube patency depending on the phase of the menstrual cycle in two groups of women: those in a follicular phase group and those in a luteal phase group. (2) Materials and Methods: The study group consisted of 25 healthy non-pregnant women aged 24 to 32. Based on the phase of the cycle confirmed in sonography, they were divided into follicular phase (FP) and luteal phase (LP) groups. The Eustachian tube and nasal cavity patency examination was carried out using a SNOT-22 Questionnaire, a rhinomanometer, and a tympanometer. (3) Results: We observed that the incidence of nasal obstruction in SNOT-22 was significantly dependent on the cycle phase (*p* = 0.012) and was lower in the FP compared to the LP. Similar relationships were noticed between the cycle phase and the rhinomanometry outcome, where the LP was associated with a lower flow. We also revealed that the incidence of ear blockage significantly depended on the cycle phase (*p* = 0.001) and was lower in the FP compared to the LP. Women whose nasal patency deteriorated during the LP also had more negative pressure values in tympanometry. We observed that patients with negative PEAK L and R levels had a lower flow in rhinomanometry. (4) Conclusions: The results highlight the menstrual cycle’s substantial impact on both subjective and objective nasal and Eustachian tube patency measurements. The novel finding in this study is that women whose nasal patency deteriorated during the luteal phase also had more negative pressure values in tympanometry. These results suggest that the deterioration of hearing during the menstrual cycle could be a result of swelling of the nasal mucosa and tubes.

## 1. Introduction

Estrogen and progesterone, hormones specific to females, undergo fluctuations during the menstrual cycle. Estrogen predominates during the follicular phase (FP), while progesterone peaks during the luteal phase (LP) (Figure 1). Estrogens produced mainly by the ovaries, released by the ovaries’ follicles, and secreted by the corpus luteum primarily regulate the growth and development of female reproductive organs and secondary sexual characteristics [1,2,3]. Progesterone, produced not only by the ovaries but also by the adrenal glands in the central nervous system, increases during the LP of the menstrual cycle, pregnancy, and embryonic development. It holds various vital roles within the body, serving as a key metabolic intermediate in synthesizing other internal steroids, including sex hormones and corticosteroids. Furthermore, it acts as a neurosteroid, playing a significant part in brain function [4,5].

These hormonal fluctuations, which are natural during menstruation, impact various physiological processes, influencing immunity, overall well-being, and sleep patterns, and may also lead to symptoms such as migraines, stress, mood swings, and anxiety. Moreover, estrogen and progesterone affect the functioning of the sense of hearing and the nasal cavity [4,6]. Estrogen serves to safeguard auditory function, potentially stimulating activity in the auditory nerve and providing neuroprotective benefits. Via their receptors in the inner ear, estrogens impact auditory signal transmission and regulate fluid electrolytes [7,8,9].

On the other hand, progesterone may interact with various steroid receptors found in the cochlea or nearby regions of the auditory system [7]. Notably, elevated progesterone levels in the LP can disturb electrolyte fluid levels, leading to sensations of ear fullness, imbalance, tinnitus, or symptoms akin to those of Meniere’s disease [10].

Additionally, it is widely acknowledged that nasal physiology can be affected by both hormones, which can be observed during the menstrual cycle when hormone levels fluctuate. While numerous studies have explored the impact of sex hormones on specific nasal functions, the decongestive response of the nasal mucosa during the menstrual cycle remains elusive [11,12,13,14]. Research confirms the presence of estrogen and progesterone receptors in the nasal mucous membrane, serous glands’ cytoplasm, and subepithelial cell nuclei. Nevertheless, data regarding receptor sensitivity to changes in estrogen and progesterone plasma concentrations during the menstrual cycle are inconclusive. Some experts suggest that female sex hormones may indirectly affect the nasal mucosa, leading to vasodilation, edema, and increased secretion by altering the expression of histamine H1 receptors and neurotransmitter concentrations such as substance P, thereby modifying nasal airflow resistance (Figure 1). Since nasal airflow determines the air volume reaching the Eustachian tube and olfactory region, changes in progesterone and estrogen plasma concentrations during the menstrual cycle may also impact the sense of hearing and smell [15,16,17]. However, there are limited data on the relationship between the menstrual cycle and nasal mucosa [18,19].

This study aimed to demonstrate changes in women’s Eustachian tube and nasal patency during different phases of the menstrual cycle in two different groups of women: those in a follicular phase group and those in a luteal phase group. This is the first report comparing the objective measurement of nasal patency with the objective result of Eustachian tube function during the menstrual cycle.

## 2. Materials and Methods

The study group consisted of 25 healthy non-pregnant women, medical students aged 24 to 32 (mean ± SD: 26.00 ± 2.58), examined at the Department of Otolaryngology in Międzylesie Specialist Hospital in Warsaw, Poland (Figure 2), and divided into two groups depending on menstrual cycle phase. This study does not include a control group because the study group consisted of healthy women in whom we examined nasal and Eustachian tube obstruction in both phases of the menstrual cycle. Each participant’s menstrual cycle was regular (mean cycle length: 28.5 ± 3 days), with ovulation. In the present study, the different phases of the menstrual cycle were decided based on the number of days from the previous menstrual cycle. Moreover, sonography was performed on day 10, 12, and 15 of the cycle to confirm the ovulatory cycle and the phase of the cycle.

Women in the FP comprised 44% (11 women) of the study group (follicular phase group—FP group), and women in the LP comprised 56% (14 women) (luteal phase group—LP group).

Patients with severe pathologies that could influence nasal patency or middle ear conditions, such as septal deviation, external nose deformations, inflammatory and cancerous lesions, and hypertrophy of the pharyngeal tonsil, were excluded from the study.

None of the women had been taking hormonal contraceptives or other drug treatments that could alter hormonal function, and they had no history of endocrine pathology.

Each patient participating in the research project underwent an ENT examination, including anterior and posterior rhinoscopy, otoscopy, and upper respiratory tract fiberoscopy. The evaluation was performed on the 3–4th day (early FP) and the 14–15th day (peri-ovulatory and early LP) of the menstrual cycle.

To evaluate changes in the openness of the nasal passages and Eustachian tubes, specialized tests called rhinomanometry and tympanometry were performed using the “COMBI 4000” ENT diagnostic module from Homoth Medizinelektronik. This device combines the functionalities of a rhinomanometer and a tympanometer. During the tympanometry test, a probe was carefully inserted into the patient’s ear canal to measure the compliance of the eardrum. Real-time monitoring of tympanic membrane compliance was displayed, allowing instant impedance assessment. The device can measure pressure values in the middle ear within a specific range. Ear canal volume was measured in milliliters, with normal values falling within a certain range. Rhinomanometry involved the insertion of nasal probes into the nostrils, allowing natural breathing while the device recorded airflow rates during inhalation and exhalation. Results were displayed for different pressure levels and sides, providing comprehensive data on nasal airflow and resistance.

Additionally, the patients were evaluated using the Sinonasal Outcome Test-22 Questionnaire (SNOT-22), a commonly employed tool for assessing the impact of sinonasal diseases on an individual’s quality of life. This questionnaire comprises 22 inquiries covering a spectrum of symptoms related to nasal and sinus issues. The questions delve into various facets, encompassing nasal and facial symptoms, disturbances in sleep patterns, and psychological well-being. The principal objective of the SNOT-22 is to offer a comprehensive appraisal of how sinonasal ailments influence daily life and overall wellness. The questionnaire items are organized into distinct domains, including rhinologic symptoms (such as nasal congestion, runny nose, and the urge to blow one’s nose), ear-related problems such as ear fullness, facial symptoms, and sleep disruptions or psychological concerns. Each question typically receives a score of 0 to 5, where 0 denotes no difficulty, and 5 denotes the most severe issue. These scores are then tallied to yield a total score, furnishing a quantitative gauge of the impact of sinonasal diseases.

The analysis was performed in R statistical software, version 4.1.2. Characteristics were presented with mean ± SD or median (IQR), depending on distribution normality. Distribution normality was verified with Shapiro–Wilk outcomes, skewness, and kurtosis. Variance homogeneity was assessed with Levene’s test. Comparisons of selected variables between SNOT outcome 0 and SNOT outcome 1–2 were analyzed with the Student’s t-test. Comparisons between cycle phases were performed with the Student’s t-test when distributions were normal or the Mann–Whitney U test for non-parametric data types. Correlation coefficients were calculated with the Spearman method due to non-normal distributions. All tests assumed alpha = 0.05.

## 3. Results

### 3.1. The Results of Sinonasal Conditions Evaluation

#### 3.1.1. The Subjective Assessment of Nasal Patency in SNOT-22

In the study group of 25 females, 10 patients (40%) reported nasal obstruction in the SNOT-22 questionnaire, among which 9 patients (90%) were in the LP group. We observed that the incidence of nasal obstruction, defined as a SNOT-22 outcome of 1 or 2, significantly depended on the cycle phase (*p* = 0.012). Incidence of nasal obstruction was lower in the FP compared to the LP (9.1%, n = 1 vs. 64.3%, n = 9), (Table 1). The severity of subjective nasal obstruction was significantly statistically correlated with the cycle phase.

#### 3.1.2. The Objective Assessment of Nasal Patency in Rhinomanometry

In this study, we observed relationships between the cycle phase and the outcome of rhinomanometry. We observed that the severity of nasal obstruction was significantly higher in women in the LP than in women in the FP (Table 1).

We observed that the level of insp was significantly higher among patients in the FP compared to patients in the LP (for 300 Pa L (*p* < 0.001), 300 Pa R (*p* = 0.002), and 150 Pa R (*p* = 0.014), respectively). The same correlation was observed in the level of exsp, which was significantly higher among patients in the FP compared to patients in the LP (for 300 Pa L (*p* < 0.001), 300 Pa R (*p* = 0.007), and 150 Pa L (*p* = 0.005), respectively). The level of the analyzed rhinomanometry parameters in the split-to-cycle phase is visualized in Figure 3.

Comparing the subjective assessment of nasal patency in SNOT-22 to the objective rhinomanometry measurements, we confirmed correlation in the LP group, where the subjective nasal obstruction was associated with a lower flow based on the outcome of rhinomanometry. This was especially observed for exsp 75Pa L (*p* = 0.029).

### 3.2. The Results of Eustachian Tube Evaluation

#### 3.2.1. The Subjective Assessment of Eustachian Tube Patency in SNOT-22

In the study group of 25 females, 9 patients (36%) reported ear blockage in the SNOT-22 questionnaire (defined as a SNOT-22 score of 1 or 2), all of whom were in the LP group, with the incidence of ear blockage significantly depending on the menstrual cycle phase (*p* = 0.001) and being lower in the FP compared to the LP (0.0% vs. 64.3%, n = 9) (Table 1).

#### 3.2.2. The Objective Assessment of Eustachian Tube Patency in Tympanometry

In this study, we observed relationships between the cycle phase and the tympanometry outcome. We confirmed a significant difference in PEAK L between patients in the FP and patients in the LP. Average PEAK L and PEAK R were positive among patients in the FP and negative among patients in the LP, with the difference of MD = 39.56 CI_95_ (*p* < 0.001) and MD = 42.87 CI_95_, (*p* < 0.001), respectively (Table 1). Levels of PEAK L and PEAK P depending on the cycle phase are visualized in Figure 4.

Additionally, statistical analysis revealed a significant correlation between the subjective and objective assessment of the Eustachian tube patency in the study group. Patients with no ear blockage had positive PEAK L and R levels, while patients with ear blockage had negative PEAK L and R levels (the difference was equal to MD = 29.92 CI_95_ *p* < 0.001 for PEAK L and MD = 32.90 CI_95_, *p* < 0.001 for PEAK R).

Data are presented as mean ± standard deviation or median (interquartile range), depending on distribution. MD is the mean or median difference (follicular phase vs. luteal phase), and CI is the confidence interval. Comparisons were run with the Student’s t-test, Mann–Whitney U test^1^, or Fisher’s exact test^2^, as appropriate.

### 3.3. The Correlation between the Results of Rhinomanometry and Tympanometry

We observed a significant correlation between the objective assessment of the Eustachian tube patency in the study group and the objective assessment of the nasal patency in rhinomanometry in the LP. Women whose nasal patency deteriorated during the LP also had more negative pressure values in tympanometry.

We observed that patients with negative PEAK L and R levels had a lower flow based on the outcome of rhinomanometry. We noticed that greater nasal obstruction correlated with lower pressures in the tympanometry examination. Spearman correlation coefficients were calculated for PEAK L and PEAK R vs. respective rhinomanometry parameters. The highest outcome was observed between PEAK R and insp 300 Pa, rho = 0.40, which indicated a positive correlation of moderate strength (Figure 5).

## 4. Discussion

While it is established that sex hormones such as estrogen and progesterone can influence nasal functions, the specific effects of these hormones on the decongestive response of the nasal mucosa during the menstrual cycle remain less clear. It is known that estrogen causes vasodilation and increases blood flow, resulting in swelling of the nasal mucosa. However, progesterone can lead to fluid retention, potentially contributing to nasal congestion but also affecting the nasal mucosa’s sensitivity and responsiveness. Finally, the menstrual phase, where both hormones’ levels are low, reduces nasal congestion. After that, in the FP, a gradual increase in estrogen is noticed, with a peak in ovulation. The next LP is initially associated with increasing progesterone and high estrogen, and then both hormones decrease if there is no pregnancy. Therefore, it could be expected that some women might experience Eustachian tube dysfunction and variations in nasal airflow and resistance during this time. Our study’s results confirmed that women in the early LP had bigger nasal obstruction problems than those in the early FP.

Moreover, we found a correlation between subjective and objective measures of nasal patency in the LP group, which was not noticed in the FP patients. Our observations are in line with the investigation by Philpott et al., which revealed that nose stuffiness happens when estrogen levels rise during ovulation in the monthly cycle [13]. Some authors compared the ovulatory and LP of the menstrual cycle and recorded nasal resistance and NOSE scores that were significantly lower in the LP than in the ovulatory, but they did not confirm a correlation between subjective and objective measures of nasal obstruction [20]. In contrast to our study, some investigators did not observe statistically important differences in nasal resistance measured using rhinomanometry between the two phases or even reported decreased airflow resistance during the peri-ovulatory phase or observed lower peak expiratory nasal flow levels during menstruation days when serum estrogen levels are at their lowest [14,21,22,23]. Therefore, the question of whether increased serum estrogen levels are responsible for nasal patency disturbance in the early LP remains a matter of debate. The presence of estrogen and progesterone receptors in the nasal mucosa adds weight to the argument that there is a clear connection between subjective and objective changes observed during periods of elevated hormone levels. One possible reason could be that estrogen receptors, when activated within the nasal mucosa, lead to the increased expression of histamine H1 receptors. This action results in vasodilation, secretion, and mucosal swelling, indirectly leading to heightened nasal resistance [17,24]. Therefore, some researchers have investigated whether medicine blocking estrogen could help clear up stuffy noses [13]. Additionally, in our study, we also found the menstrual cycle’s impact on both subjective and objective measures of Eustachian tube patency. We noticed that the incidence of ear blockage significantly depended on the cycle phase and was observed only in the LP group.

Furthermore, we revealed a significant correlation between the subjective and objective assessment of the Eustachian tube patency. There is limited research on tympanometry results that show changes in the menstrual cycle, but some investigators have noticed a moderate decline in Eustachian tube function associated with elevated blood levels of estradiol. They observed a correlation between estrogen levels and Eustachian tube function across a broad spectrum of estrogen concentrations in women undergoing the induction of ovulation. Other authors have previously reported an association between menstrual phase and audiometry threshold for selective frequencies [25]. They found that the audiometric threshold was worse in the FP at 4000 Hz, and during ovulation, the threshold was better than that of any other phases at 1000 Hz. Simultaneously, they noticed a correlation between the menstrual cycle phase and the amplitude of DPOAE [25]. Emami et al. pointed out that alterations in ovarian hormones may trigger variations in hearing for certain women, and heightened progesterone levels during the LP could result in irregularities in auditory function [26]. Few authors acknowledge that the hearing sensitivity fluctuations during the menstrual cycle is a consequence of cochlear receptors for both estrogen and progesterone, which influence inner ear balance and auditory capabilities [7,27].

It has to be mentioned that our study is the first report to objectively compare the impact of nasal patency on Eustachian tube function during the menstrual cycle. The novel finding in this study is that women whose nasal patency deteriorated during the LP also had more negative pressure values in tympanometry. These results suggest that the deterioration of hearing during the menstrual cycle could be a result of swelling of the nasal mucosa and tubes.

The next strength of our study is that we made both subjective and objective measurements of nasal patency and Eustachian tube function and compared the results from two diagnostic tools. On the other hand, the limitation of the study was that we did not observe the parameters in two phases of the menstrual cycle in the same individual, which could be the subject of the next research, because it is a pilot study. Additionally, the SNOT-22 questionnaire pertains to symptoms currently being observed, so it would be valuable to compare how it changes for each individual patient in each phase of the cycle. Although we had a limited number of participants in our study, our findings indicate the necessity to understand the connection between the menstrual cycle and both the Eustachian tube and upper respiratory tract function.

## 5. Conclusions

The clinical significance of these results is the knowledge that, in some women, Eustachian tube dysfunction and nasal patency resistance may manifest in the LP; understanding this relationship better could improve the management of nasal congestion in women experiencing cyclic symptoms. This is a pilot study with some limitations, so a larger sample and an analysis of the same patient in different phases of the menstrual cycle are necessary for significant results.

## Figures and Tables

**Figure 1 diagnostics-14-02044-f001:**
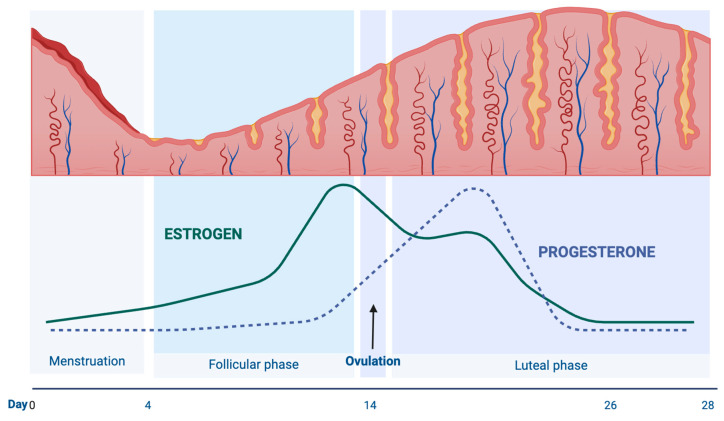
Fluctuations in female sex hormones and the membrane mucosa during the menstrual cycle.

**Figure 2 diagnostics-14-02044-f002:**
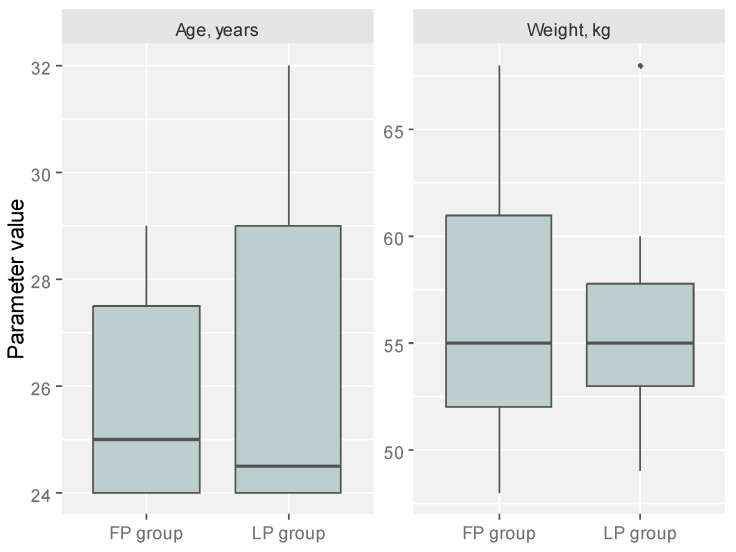
Boxplots present the age and weight of analyzed patients in the split-to-cycle phase. FP—follicular phase, LP—luteal phase.

**Figure 3 diagnostics-14-02044-f003:**
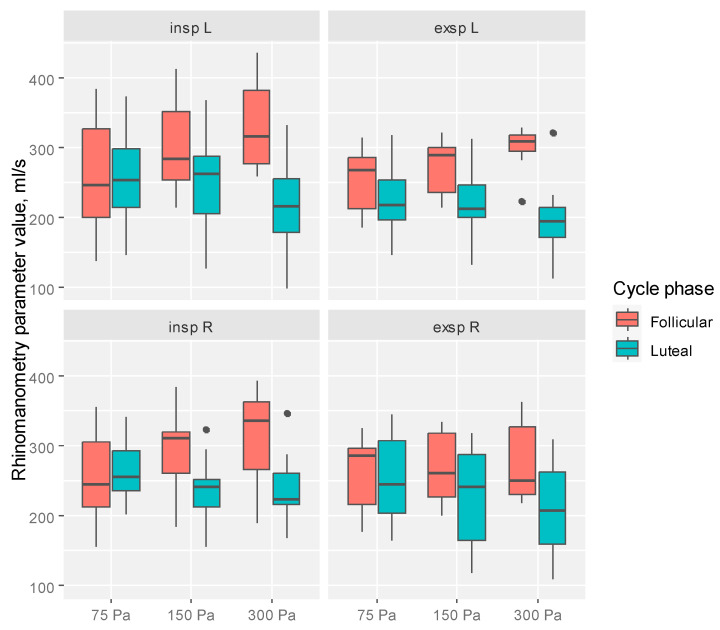
Boxplots presenting the level of the analyzed rhinomanometry parameters in the split-to-cycle phase.

**Figure 4 diagnostics-14-02044-f004:**
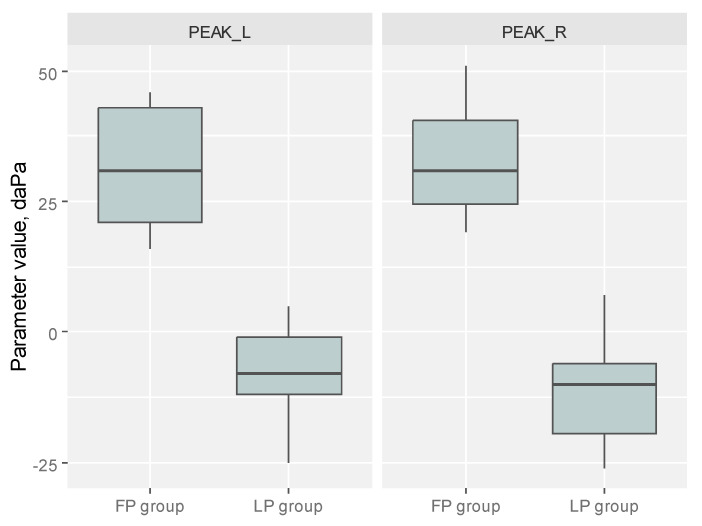
Boxplots present the level of the analyzed tympanometry parameters depending on the cycle phase in the study group. FP—follicular phase, LP—luteal phase.

**Figure 5 diagnostics-14-02044-f005:**
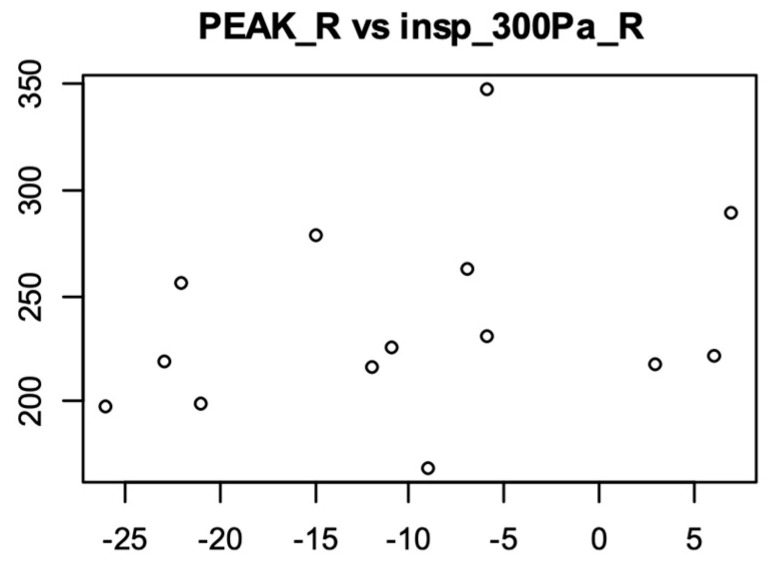
Correlation analysis of PEAK L/PEAK R with rhinomanometry parameters, among women in the luteal phase.

**Table 1 diagnostics-14-02044-t001:** Results of SNOT −22, tympanometry, and rhinomanometry in the study group. Comparisons run with *t*-Student test, Mann-Whitney U test ^1^ or Fisher exact test ^2^, as appropriate.

Variable	Follicular Phase (n = 11)	Luteal Phase (n = 14)	MD (95% CI)	*p*
SNOT-22				
Nasal obstruction				
0	10 (90.9)	5 (35.7)	-	0.012 ^2^
1–2	1 (9.1)	9 (64.3)		
Ear blockage				
0	11 (100.0)	5 (35.7)	-	0.001 ^2^
1–2	0 (0.0)	9 (64.3)		
Tympanometry				
PEAK L, daPa	31.64 ± 11.89	−7.93 ± 8.94	39.56 (30.96;48.17)	<0.001
PEAK R, daPa	32.73 ± 10.08	−10.14 ± 10.65	42.87 (34.19;51.55)	<0.001
Rhinomanometry				
insp 75 Pa L, mL/s	264.27 ± 82.53	254.07 ± 62.53	10.20 (−49.74;70.14)	0.728
insp 150 Pa L, mL/s	301.91 ± 68.22	246.79 ± 65.76	55.12 (−0.59;110.84)	0.052
insp 300 Pa L, mL/s	328.64 ± 63.96	217.14 ± 67.13	111.49 (56.68;166.31)	< 0.001
exsp 75 Pa L, mL/s	252.91 ± 46.39	226.57 ± 48.06	26.34 (−13.12;65.80)	0.181
exsp 150 Pa L, mL/s	270.55 ± 38.71	219.07 ± 42.57	51.47 (17.36;85.59)	0.005
exsp 300 Pa L, mL/s	308.00 (293.50;318.00)	194.00 (171.00;213.75)	114.00 (86.00;137.00)	< 0.001 ^1^
insp 75 Pa R, mL/s	256.45 ± 62.19	263.57 ± 38.72	−7.12 (−49.03;34.80)	0.729
insp 150 Pa R, mL/s	291.64 ± 57.81	239.29 ± 40.74	52.35 (11.59;93.11)	0.014
insp 300 Pa R, mL/s	315.18 ± 66.95	237.79 ± 45.53	77.40 (30.84;123.96)	0.002
exsp 75 Pa R, mL/s	259.82 ± 49.84	252.93 ± 55.89	6.89 (−37.57;51.35)	0.751
exsp 150 Pa R, mL/s	269.45 ± 48.75	227.71 ± 69.23	41.74 (−9.25;92.73)	0.104
exsp 300 Pa R, mL/s	278.00 ± 53.06	207.43 ± 63.74	70.57 (21.12;120.03)	0.007

## Data Availability

The original contributions presented in the study are included in the article, further inquiries can be directed to the corresponding author/s.

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
