# Peer review of "Evaluation of Changes in the Patency of the Nasal Cavity and Eustachian Tube Depending on the Phase of the Menstrual Cycle: A Pilot Study"

_diagnostics, 2024, doi:10.3390/diagnostics14182044_

Round 1

Reviewer 1 Report

Comments and Suggestions for Authors

The authors wrote an article about Evaluation of Changes in the Patency of the Nasal Cavity and  Eustachian Tube Depending on the Phase of the Menstrual Cycle. The article is very interesting, the topic is hot. I have some suggestion to improve the quality of the manuscript to give it a better impact in literarute. 

1. In the method, The exclusion criteria must specify which ENT pathologies must be excluded, for example also a frequent septal deviation, then the study is on different groups of patients and there could be several biases.

2. In the statistical analysis you wrote that "Comparisons of selected variables between SNOT outcome 0 and 131 SNOT outcome 1-2 were analyzed with the t-Student test." For the rest of the variables, what tests did you use and for the non-parametric variables?

3. Underline very well the limitation that you did not observe the parameters in two phases of the menstrual cycle in the same individual (introductio; aim; abstract; conclusion)

4. Regarding the pressure values of the timpanometry the mean of 31-32 dB in follicular phase is too positive, usually a normal pressure is abou -10 and +5. Do you use a different calibration?

5. In the conclusion, underline that this is a preliminary study with many limitations and that a larger sample and an analysis on the same patient are necessary for significant result.

Author Response

Reviewer’s comment: In the method, The exclusion criteria must specify which ENT pathologies must be excluded, for example also a frequent septal deviation, then the study is on different groups of patients and there could be several biases.

Answer: Thank you for this comment. We have made the proposed correction and added information in  the method section (text marked in green).

Reviewer’s comment: In the statistical analysis you wrote that "Comparisons of selected variables between SNOT outcome 0 and SNOT outcome 1-2 were analyzed with the t-Student test." For the rest of the variables, what tests did you use and for the non-parametric variables?

Answer: Thank you for this comment.

The analysis was performed in R statistical software, version 4.1.2. Characteristics were presented with mean ± SD or median (IQR), depending on distribution normality. Distribution normality was verified with Shapiro-Wilk outcomes, skewness and kurtosis. Variance homogeneity was assessed with Levene’s test. Comparisons of selected variables between SNOT outcome 0 and SNOT outcome 1-2 were analyzed with the t-Student test. Comparisons between cycle phases were performed with the t-Student test when distributions were normal or Mann-Whitney U test for non-parametric data type. Correlation coefficients were calculated with the Spearman method due to non-normal distributions. All tests assumed alpha = 0.05.

I mentioned that information in Method section (text marked in green) and I added an explanation to Table 1 (text marked in green).

Reviewer’s comment: Underline very well the limitation that you did not observe the parameters in two phases of the menstrual cycle in the same individual (introduction; aim; abstract; conclusion)

Answer: Thank you for this comment. We have made the proposed correction (the text marked in green in all sections mentioned above and the Method section).

Reviewer’s comment: Regarding the pressure values of the tympanometry the mean of 31-32 dB in follicular phase is too positive, usually a normal pressure is about-10 and +5. Do you use a different calibration?

Answer: Thank you for this comment. We used calibration according to the manual. Information provided in our device manual is as follow: “A middle ear pressure of +/- 500 to 100 daPa is considered as normal. The device operates over the pressure range of +200 to -400 daPa”.

Reviewer’s comment: In the conclusion, underline that this is a preliminary study with many limitations and that a larger sample and an analysis on the same patient are necessary for significant result.

Answer: Thank you for this comment. We have made the proposed correction (text marked in green).

Reviewer 2 Report

Comments and Suggestions for Authors

Figure 1 is a representative picture of changes in the endometrium. Is it right?

SNOT-22 is an evaluation questionnaire about subjective nasal symptoms over the past two weeks. I wonder what we can think about this.

There is absolutely no control or analysis of participants. There is no mention of why the patient was brought to the hospital, or any anatomical problems or allergic status. In this situation, I wonder if subjective and objective evaluation based solely on the menstrual cycle is meaningful.

Author Response

Reviewer’s comment: Figure 1 is a representative picture of changes in the endometrium. Is it right?

Answer: Thank you for this comment. The changes occurring during the phases of the menstrual cycle are similar in all mucous membranes, so to be more precise  we have changed the text under the picture- “Fluctuations in female sex hormones and membrane mucosa during the menstrual cycle”.

Reviewer’s comment: SNOT-22 is an evaluation questionnaire about subjective nasal symptoms over the past two weeks. I wonder what we can think about this.

Answer: Thank you for this valuable comment. We have made the proposed correction. Additionally, the SNOT-22 questionnaire pertains to symptoms currently being observed, so it would be valuable to compare how it changes for each individual patient in each phase of the cycle- we have added this information to the discussion (the text marked in green).

Reviewer’s comment: There is absolutely no control or analysis of participants. There is no mention of why the patient was brought to the hospital, or any anatomical problems or allergic status. In this situation, I wonder if subjective and objective evaluation based solely on the menstrual cycle is meaningful.

Answer: Thank you for this notice. The female students were at the hospital because they had otolaryngology classes. The study group consisted of 25 healthy non-pregnant women, medical students aged 24 to 32 examined at the Department of Otolaryngology in Międzylesie Specialist Hospital in Warsaw, Poland. We underlined this information in the Material and Method section.

Reviewer 3 Report

Comments and Suggestions for Authors

Re: 

The present study is an attempt to evaluate changes in patency of the nasal cavity and the Eustachian tube in relation to the follicular (FP) and the luteal phases (LP) of the menstrual cycle.

25 non-pregnant women, aged 24 to 32 years, were recruited for the study. Their phase of the menstrual cycle was determined by sonography. 11 of the woman were in the FP and 14 in the LP group. After a conventional ENT investigation the women were evaluated by rhinomanometry and tympanometry. Furthermore they were subjected to questionnaire – SNOT-22 – containing 22 inquiries related to nasal and sinus issues as well as psychological well being.

The rhinomanometry showed a low flow associated with LP and thus a higher severity of nasal obstruction. Tympanometry revealed ear blockage for the LP group. Regarding the SNOT-22 analysis 10 patients noted nasal obstruction of which 9 belonged to the LP group. 

9 women reported ear blockage and all om them belonged to the LP group.

The authors conclude that the nasal patency and the patency of the Eustachian tube deteriorated in the LP. They also suggest that the deterioration of hearing during the menstrual cycle is mostly a result of swelling of the nasal mucosa and tubes rather than an impact on cochlear receptors for hormones.

The results are interesting, and the study quite well performed. Nevertheless there are some points which have to be considered before I am ready to recommend this paper for publication in Diagnostics. See questions/comment below.

Questions/comments

Abstract – lines 30-32 – I can´t agree with the suggestion “that the deterioration of hearing during the menstrual cycle is mostly a result of swelling of the nasal mucosa and tubes rather than an impact on cochlear receptors for hormones”. Without audiological tests in the FP and LP groups we have no objective measures of the hearing levels. Why didn´t the authors also include hearing tests in their study?

Material and methods – lines 90-91 – Women in FP comprised 44% of the study group- and women in LP comprised 56%.  The reader would prefer number of women in figures 11 and 14 respectively instead of percentage!

Results – lines142-143 – “ In our group of 14 females in the LP, 9 women (64%) reported nasal obstruction in SNOT-22 questionnaire and only one in the FP.”  This sentence can be omitted as it is already told above in lines 136-137.

Results – lines147-148 – “The LP was associated with lower flow.  We observed that the severity of nasal obstruction was significantly higher in women in LP than in women with FP.” This is another example where two sentences contain the same information and where the first can be omitted.

Results – lines 166-170 – These 2 sentences almost contain the same information and could be reduced to 1 sentence.

Results – lines 193-196 – “We observed a significant correlation between the objective assessment of the Eustachian tube patency in the study group and the objective assessment of the nasal patency in rhinomanometry in LP.”  I cannot find any data om how this significant correlation was calculated.

Discussion – lines 259-263 – Part of the Discussion is speculative and I don´t agree with the statement  -  “The novel finding in this study is that women whose nasal patency deteriorated during LP also had more negative pressure values in tympanometry. These results suggest that the deterioration of hearing during the menstrual cycle is mostly a result of swelling of the nasal mucosa and tubes than an impact on cochlear receptors for hormones.” I cannot see how the present results suggest that the deterioration of hearing during the menstrual cycle is mostly a result of swelling of the nasal mucosa and tubes.

 The present study is an attempt to evaluate changes in patency of the nasal cavity and the Eustachian tube in relation to the follicular (FP) and the luteal phases (LP) of the menstrual cycle.

25 non-pregnant women, aged 24 to 32 years, were recruited for the study. Their phase of the menstrual cycle was determined by sonography. 11 of the woman were in the FP and 14 in the LP group. After a conventional ENT investigation the women were evaluated by rhinomanometry and tympanometry. Furthermore they were subjected to questionnaire – SNOT-22 – containing 22 inquiries related to nasal and sinus issues as well as psychological well being.

The rhinomanometry showed a low flow associated with LP and thus a higher severity of nasal obstruction. Tympanometry revealed ear blockage for the LP group. Regarding the SNOT-22 analysis 10 patients noted nasal obstruction of which 9 belonged to the LP group. 

9 women reported ear blockage and all om them belonged to the LP group.

The authors conclude that the nasal patency and the patency of the Eustachian tube deteriorated in the LP. They also suggest that the deterioration of hearing during the menstrual cycle is mostly a result of swelling of the nasal mucosa and tubes rather than an impact on cochlear receptors for hormones.

The results are interesting, and the study quite well performed. Nevertheless there are some points which have to be considered before I am ready to recommend this paper for publication in Diagnostics. See questions/comment below.

Questions/comments

Abstract – lines 30-32 – I can´t agree with the suggestion “that the deterioration of hearing during the menstrual cycle is mostly a result of swelling of the nasal mucosa and tubes rather than an impact on cochlear receptors for hormones”. Without audiological tests in the FP and LP groups we have no objective measures of the hearing levels. Why didn´t the authors also include hearing tests in their study?

Material and methods – lines 90-91 – Women in FP comprised 44% of the study group- and women in LP comprised 56%.  The reader would prefer number of women in figures 11 and 14 respectively instead of percentage!

Results – lines142-143 – “ In our group of 14 females in the LP, 9 women (64%) reported nasal obstruction in SNOT-22 questionnaire and only one in the FP.”  This sentence can be omitted as it is already told above in lines 136-137.

Results – lines147-148 – “The LP was associated with lower flow.  We observed that the severity of nasal obstruction was significantly higher in women in LP than in women with FP.” This is another example where two sentences contain the same information and where the first can be omitted.

Results – lines 166-170 – These 2 sentences almost contain the same information and could be reduced to 1 sentence.

Results – lines 193-196 – “We observed a significant correlation between the objective assessment of the Eustachian tube patency in the study group and the objective assessment of the nasal patency in rhinomanometry in LP.”  I cannot find any data om how this significant correlation was calculated.

Discussion – lines 259-263 – Part of the Discussion is speculative and I don´t agree with the statement  -  “The novel finding in this study is that women whose nasal patency deteriorated during LP also had more negative pressure values in tympanometry. These results suggest that the deterioration of hearing during the menstrual cycle is mostly a result of swelling of the nasal mucosa and tubes than an impact on cochlear receptors for hormones.” I cannot see how the present results suggest that the deterioration of hearing during the menstrual cycle is mostly a result of swelling of the nasal mucosa and tubes.

Author Response

Reviewer’s comment: Abstract – lines 30-32 – I can´t agree with the suggestion “that the deterioration of hearing during the menstrual cycle is mostly a result of swelling of the nasal mucosa and tubes rather than an impact on cochlear receptors for hormones”. Without audiological tests in the FP and LP groups we have no objective measures of the hearing levels. Why didn´t the authors also include hearing tests in their study?

Answer: Thank you for this comment. We agree that these were exaggerated conclusions. We have revised this information in both the discussion and the abstract.(the text marked in green)

Reviewer’s comment: Material and methods – lines 90-91 – Women in FP comprised 44% of the study group- and women in LP comprised 56%.  The reader would prefer number of women in figures 11 and 14 respectively instead of percentage!

Answer: Thank you for this opinion. We have made the proposed correction (We placed the numbers in parentheses- the text marked in green).

Reviewer’s comment:

Results – lines142-143 – “ In our group of 14 females in the LP, 9 women (64%) reported nasal obstruction in SNOT-22 questionnaire and only one in the FP.”  This sentence can be omitted as it is already told above in lines 136-137.

Results – lines147-148 – “The LP was associated with lower flow.  We observed that the severity of nasal obstruction was significantly higher in women in LP than in women with FP.” This is another example where two sentences contain the same information and where the first can be omitted.

Results – lines 166-170 – These 2 sentences almost contain the same information and could be reduced to 1 sentence.

Answer: Thank you for those comments. We have made the proposed corrections as suggested.

Reviewer’s comment: Results – lines 193-196 – “We observed a significant correlation between the objective assessment of the Eustachian tube patency in the study group and the objective assessment of the nasal patency in rhinomanometry in LP.”  I cannot find any data om how this significant correlation was calculated.

Answer: Thank you for this valuable comment. We have made the correction, added new figure (figure 5) to presence  this information- Spearman correlation coefficients were calculated for PEAK L and PEAK R vs respective rhinomanometry parameters. The highest outcome was observed between PEAK R and insp 300 Pa, rho = 0.40, which indicated a positive correlation of moderate strength.

Reviewer’s comment: Discussion – lines 259-263 – Part of the Discussion is speculative and I don´t agree with the statement  -  “The novel finding in this study is that women whose nasal patency deteriorated during LP also had more negative pressure values in tympanometry. These results suggest that the deterioration of hearing during the menstrual cycle is mostly a result of swelling of the nasal mucosa and tubes than an impact on cochlear receptors for hormones.” I cannot see how the present results suggest that the deterioration of hearing during the menstrual cycle is mostly a result of swelling of the nasal mucosa and tubes.

Answer: Thank you for this opinion. We have made the proposed correction. We agree that these were exaggerated conclusions. We have revised this information in both the discussion and the abstract.

Round 2

Reviewer 3 Report

Comments and Suggestions for Authors

The authors have responded well to my comments/criticism

Author Response

Thank you for this comment.